# Transcription Factor *VM1G_06867*: A Requirement for Growth, Pathogenicity, Development, and Maintenance of Cell Wall Integrity in *Valsa mali*

**DOI:** 10.3390/jof9060692

**Published:** 2023-06-20

**Authors:** Yufei Diao, Jiyang Jin, Xiong Xiong, Chengming Yu, Yehan Tian, Duochuan Li, Huixiang Liu

**Affiliations:** 1Shandong Research Center for Forestry Harmful Biological Control Engineering and Technology, College of Plant Protection, Shandong Agricultural University, Tai’an 271018, China; f18853886856@163.com (Y.D.);; 2Mountain Tai Forest Ecosystem Research Station of State Forestry Administration, Forestry College, Shandong Agricultural University, Tai’an 271018, China

**Keywords:** *Valsa mali*, *VmSom1*, *VM1G_06867*, growth, conidiation, pathogenicity, cell wall integrity

## Abstract

Apple canker disease, caused by *Valsa mali*, is one of the most serious apple tree diseases in China. *VmSom1* is an important transcription factor that acts on the cyclic adenosine signaling pathway (cAMP/PKA), regulating the growth, development, morphological differentiation, and pathogenic forces of the pathogen. We perform transcriptome analysis of the *VmSom1* deletion mutant and the wild-type strain 11-175 and identify a significantly differentially expressed gene, *VM1G_06867*, a zinc finger motif transcription factor in *V. mali*. In this study, we obtain the *VM1G_06867* gene using the single deletion mutant via homologous recombination. To determine the relationship between *VmSom1* and *VM1G_06867*, we also obtain a double deletion mutant *ΔVmSom1/06867*. Compared to the wild-type strain 11-175, the single deletion mutant *VM1G_06867* shows a drastic reduction in growth rate and forms more pycnidia on the PDA medium. Additionally, the growth of the mutant is inhibited by SDS, Congo red, and fluorescent brighteners. In comparison to the single deletion mutant *VmSom1*, the double deletion mutant *ΔVmSom1/06867* shows no significant change in growth or conidiation and is unable to produce conidia. The growth rate is significantly increased in Congo red, NaCl, and Sorbitol mediums. These results demonstrate that *VM1G_06867* plays important roles in growth, pathogenicity, asexual development, and maintenance of cell wall integrity. *VM1G_06867* can recover osmotic stress and cell wall integrity defects caused by the deletion of *VmSom1*, as well as restore the loss of pathogenicity caused by the deletion of the *VmSom1* gene, but not completely.

## 1. Introduction

Apple canker disease, caused by *Valsa mali*, is a serious apple tree disease in China, leading to tree death and orchard destruction. The disease mainly occurs in apple-producing areas of East Asia [1,2].

A transcription factor is a special structure protein that regulates the initiation of gene transcription by binding to the upstream promoter sequence of the target gene [3]. Transcription factors interact with elements that regulate the gene’s promoter region, playing a crucial role in physiological function and cell death [4,5]. Class Zn2Cys6 transcription factors, also known as C6, Zn2C6, zinc bi-nuclear cluster, and zinc cluster, are unique to fungi [6]. The study of *Alternaria brasicola* reveals that Zn2Cys6-type transcription factors affected fungal pathogenicity, and knocking down transcription factors containing two Zn2Cys6 domains and one transmembrane domain resulted in the loss of pathogenicity [7]. The Zn2Cys6 transcription factor *EBR1* is also a fungal-unique transcription factor, and its knockout in *Fusarium graminearum* PH-1 leads to decreased polar growth and lower pathogenicity [8]. Through the study of 104 fungal-unique Zn2Cys6 transcription factors, it is found that 61 of them are necessary for the growth and development of the fungus. Furthermore, these transcription factors function in multiple stages of growth and development and also regulate the pathogenicity of *Magnaporthe grisea* [9]. In Sordaria macrospola, the Zn2Cys6 class transcription factor *Pro1* plays a crucial role in conidial maturation, and the knockout of *ProA* leads to sexual reproduction [10]. Additionally, fungal-unique Zn2cys6 transcription factors play an important role in nutrient utilization during pathogen infection and secondary metabolism. For example, the specific transcription factor *hmgR* of *Penicillium marneffei* regulates the expression of genes related to tyrosine synthesis, thereby affecting the yield of melanin precursors [11]. All the studies mentioned above demonstrate that unique fungal transcription factors regulate the morphology and development of the pathogen and play a significant role in the pathogen’s infection process. However, there have been few studies on transcription factors in the apple canker pathogen to date.

*Som1* (cAMP-dependent protein kinase pathway protein) is an important transcription factor located downstream of the cAMP-PKA signaling pathway [12]. In *Verticillium dahliae*, *Som1* controls adhesion, the oxidative stress response, and the developmental genetic networks required for conidia, microsclerotia formation, and pathogenicity [13]. In *Magnaporthe oryzae*, *Som1* is essential for spore and attachment production and plays a role in cell wall differentiation, regulation of melanin deposition, and surface hydrophobicity [14]. In *Aspergillus fumigatus*, the absence of *Som1* leads to slow growth of mycelia and blocked asexual development and only undifferentiated aerial mycelia can be formed [15]. The lack of *Som1* in *Metarhizium acridum* lowers conidial production, delays conidial germination, and weakens heat and UV-B tolerance [16]. Previously, *Som1* had not been studied in apple putrefaction.

Previous studies conducting transcriptome analyses found an upregulated and downregulated series of differentially expressed transcription factors in *V. mali* (unpublished). In this study, we perform double-join PCR and gap-repair techniques to generate the deletion mutant *ΔVM1G_06867* and the complemented strain *ΔVM1G_06867ct*. In addition to affecting growth, *VM1G_06867* is involved in conidiation, osmotic stress response, and maintenance of cell wall integrity in *V. mali*.

## 2. Materials and Methods

### 2.1. Strains and Culture Conditions

The wild-type strain sdau11-175 is provided by the Research Laboratory of Forest Pathogen and Host Molecular Interactions at the College of Shandong Agricultural University. All strains in this study are cultured in potato dextrose agar (PDA) medium (per 1000 mL medium, 200 g potato extract, 20 g dextrose, and 1.5 g/mL agar).

The growth rate of the wild-type strain, deletion mutant, and supplementation strains on PDA media is assayed by measuring colony diameter three days post-inoculation (dpi) at 25 °C. TB3 media containing hygromycin B (per 1000 mL, 3 g yeast extract, 3 g casamino acids, 200 g sucrose, and 1.5 g/mL agar) is used to select resistant transformants in the process of gene deletion or complementation. NaCl (0.5 M), sorbitol (0.5 M, 1M), Congo red (CR, 300 μg/mL), fluorescent whitening agent (CFW, 200 μg/mL), and sodium lauryl sulfate (SDS, 0.005%, 0.01%) are added to PDA for stress response assays. The carbon and nitrogen sources are analyzed with Czapek’s medium, and the required nutrient elements are determined by changing different carbon and nitrogen sources.

### 2.2. Bioinformatic Analysis

The *VM1G_06867* protein was downloaded from the whole genomic sequence of *Valsa mali* [17]. Homologous sequences of other fungi were downloaded from the GenBank database (http://www.ncbi.nlm.nih.gov, accessed on 10 April 2023). The *VM1G_06867* protein sequences were analyzed using the ClustalW2 tool, and the phylogenetic tree was constructed using MEGA 7.0 [18]. The conserved domains are predicted using the Conserved Domain Search Service (NCBI).

### 2.3. Identification, VM1G_06867 Gene Replacement, and Complementation

The *VM1G_06867* deletion mutant is constructed by replacing the open reading frame (ORF) of *VM1G_06867* gene with the hygromycin b phosphotransferase gene (hph) gene [19]. Primer pairs 06867-SY-F/R and 06867-XY-F/R are used to amplify the 1 kb upstream and 1 kb downstream flanking sequences, respectively, while primer pairs HPH-F and HPH-R are used to amplify hph. Meanwhile, the replacement fragment is constructed using double-joint PCR, and the polyethylene glycol (PEG) method is used to transform the *VM1G_06867* replacement fragment into 11-175 protoplasts [20]. The confirmation of the *VM1G_06867* deletion mutant *ΔVM1G_06867* is performed via PCR with four primer pairs (Appendix A) as described.

The complementation of the deletion mutant is generated using the gap repair approach. *VM1G_06867* fragments with their promoters (about 2.5 kb) are generated using primer pairs 06867-HB-F/R. The XhoI-digested plasmid pFL2 and the amplified fragment are co-transformed into the yeast strain XK1-25. The resulting constructs are then transformed into the protoplasts of the *VM1G_06867* deletion mutant [21]. The geneticin-resistant transformants *ΔVM1G_06867-C* are confirmed using PCR with primer pair 06867-HB-F/R (Appendix A).

### 2.4. Double Deletion Genes VM1G 06867 and VmSom1

The double deletion method is the same as the single deletion, with some modifications. The PEG method is used to transform the *VM1G_06867* replacement fragment into protoplasts of the *VmSom1* deletion mutant. The geneticin-resistant transformants are confirmed via PCR. The double deletion mutant of *VM1G_06867* and *VmSom1* was obtained and named *ΔVmSom1/06867* (Appendix A).

### 2.5. Growth Rate Determination

To evaluate the roles of *VM1G_06867* in *V. mali*, we measure the wild type, single deletion mutant, and double deletion mutant mycelial growth on PDA medium. After three days of culture, we measure the colony diameter. For each strain, the experiment is repeated ten times, and the experiment itself is repeated three times.

### 2.6. Conidial Production and Germination

To determine whether *VM1G_06867* is involved in conidiation, we count the number of pycnidia in the unit colony area of different strains. To further assess whether *VM1G_06867* affects conidia germination, we coat the conidia on PDA media. We repeat the experiment ten times for each strain, and the experiment itself is repeated three times.

### 2.7. Stress Sensitivity Assay

We add the osmotic stress factors of NaCl (0.5 M) and sorbitol (0.5 M) to the PDA medium to assay whether the gene regulates the osmotic stress of the pathogen. We add cell wall interferer factors of Congo red (300 µg/mL) and fluorescent brightener (200 µg/mL) to the PDA medium to assay the sensitivity of the mutant to cell wall interferers. Moreover, we add the cell-membrane-damaging agent SDS (0.01%) to the PDA medium to assay the sensitivity of the mutant to cell membrane interferers. We determine the colony after three days of inoculation. Each treatment is repeated ten times, and the experiment is repeated three times.

### 2.8. Analysis of the Effect of Different Carbon-Nitrogen on the Growth of Gene VM1G_06867

To make media with different carbon sources, we replace sucrose with lactose, trehalose, fructose, maltose, and glucose of equal quality. We prepare media from different nitrogen sources with equal mass: peptone, ammonium chloride, urea, sodium nitrate, potassium nitrate, and ammonium sulfate. We determine the colony size after three days. Each treatment is repeated ten times, and the experiment is repeated three times.

### 2.9. Infection Assays on Apple Fruit and Twigs

For assaying virulence on apple fruit and twigs, we culture the wild type, single deletion mutant, and double deletion mutant on PDA for three days. We take 5 mm agar plugs from the edge of a colony and inoculate armature and scald wounds on fruit and twigs of Malus domestica borkh. ‘Fuji’ inoculated tissues are incubated at 25 °C for nine days, and the length of the lesions is recorded. Each treatment is repeated ten times, and the experiment is repeated three times.

### 2.10. Statistical Analysis

We use Fisher’s least significant difference (LSD) in the SPSS software package for statistical analysis (*p* < 0.05).

## 3. Results

### 3.1. Bioinformatical Characterizations of VM1G_06867 in V. mali

We identify the single copy *VM1G_06867* gene (Accession number KUI71623.1) in the genome of *V. mali*. The gene sequence is 3228 bp and encodes an 860 amino acid (AA) protein. The VM1G_06867 protein contains a GAL4 super family domain at 34–71AA. Additionally, a fungal_TF_MHR super family domain is predicted at amino acids 373–821 of VM1G_06867 protein. (Figure 1A). A phylogenetic tree was constructed revealing VM1G_06867 showed high similarity with orthologous genes of other fungi. VP1G_00202 showed 95.58% similarity, VMCG_02875 showed 90.82% similarity, VSDG_01918 had 90.01% similarity, VPNG_09881 has 86.86% similarity, CSIMO1_09379 showed 65.43% similarity, the BEA4 similarity was 65.97%, and the CTA1_4410 similarity was 65.29%. The VM1G_06867 had a 100% identity with its homolog in the industrially important *Valsa mali var. Pyri* (Figure 1B).

### 3.2. Role of VM1G_06867 in Growth

To investigate the function of *VM1G_06867* in *V. mali*, we use the hygromycin b phosphotransferase gene (hph) to replace the entire ORF of *VM1G_06867* to generate the *VM1G_06867* deletion mutant. The detection of *VM1G_06867* shows that an expected band is found in the deletion mutants. The detection of complementation in the *VM1G_06867* deletion mutant shows that an expected band is found in the complemented mutants. Double deletion genes are obtained in the same way (Appendix A).

To assay the roles of *VM1G_06867* in *V. mali*, we inoculate the wild type and deletion mutants on a PDA medium. The radial growth rate of the *VM1G_06867* mutant is reduced by approximately 13% compared to the wild type. The growth rate of the complemented mutant is restored to that of the wild type. The double deletion mutant *ΔVmSom1/06867* shows the same characteristics as the single deletion mutant *VmSom1*, and there is no significant difference in the growth rate between them. Compared with the wild type, the aerial mycelia are reduced, the colony color is white, and the growth rate of mycelia is significantly reduced (Figure 2). These results show that *VM1G_06867* plays an important role in the growth of *V. mali*.

### 3.3. Negative Regulation of Conidiation and Impact on Conidial Germination by VM1G_06867

To evaluate whether *VM1G_06867* is involved in conidiation, we measure the number of pycnidia in the unit colony area of wild-type and deletion mutant strains. The deletion mutant strain produces more pycnidia than the wild type on PDA at 20 dpi. The previously observed phenomenon is reversed when the *VM1G_06867* gene is reintroduced into the deletion mutant strain (Figure 3A). This result indicates that *VM1G_06867* is a negative regulator of conidiation. To further evaluate whether *VM1G_06867* affects conidial germination, we collect conidia and observe the spore germination rate using the suspension drop method. After 18 h post-inoculation (hpi), the conidia swell readily and become ellipsoidal or globular in shape. At 24 hpi, conidia typically germinate and produce one to two germ tubes from both ends of a conidium in the wild type, and the hyphae of the wild type are longer than those of the *VM1G_06867* deletion mutant. However, the conidia of the deletion mutant are able to develop into mature hyphae (Figure 3B). This result indicates that the deletion of *VM1G_06867* slows conidial germination.

### 3.4. Stress Influence of VM1G_06867 on Response to Osmotic Stress and Cell-Wall Integrity Inhibitor

To test whether *VM1G_06867* plays a role in the response to abiotic stress, we inoculate the wild type, *VM1G_06867* single deletion mutant, complemented strain, and double deletion mutant on PDA supplemented with NaCl, sorbitol (osmotic stress), Congo red, fluorescent brightener (cell wall inhibitor), tebuconazole (germicide), and SDS (cell membrane damaging agent). Results show that the growth inhibitions of fluorescent brightener, NaCl, SDS, sorbitol, and Congo red on the *VM1G_06867* deletion mutant are higher than those on the wild-type and the complemented mutant strains. However, the inhibition rate of the *VM1G_06867* deletion mutant is not significantly different from the wild-type strain on PDA supplemented with tebuconazole (Figure 4). These results show that *VM1G_06867* regulates *V. mali*’s response to osmotic stress and maintenance of cell wall integrity.

Subsequently, the gene double deletion mutant *ΔVmSom1/06867* is cultured in a medium containing NaCl (0.5 M), Sorbitol (0.5 M), Congo red (400 µg/mL), fluorescent brightener (400 µg/mL), and SDS (0.01%). The growth rate of mycelium is significantly lower than that of the wild-type strain. Then, the double deletion mutant *ΔVmSom1/06867* is compared with the *VmSom1* single deletion mutant. On a medium containing Congo red, NaCl, and Sorbitol, the growth rate increases significantly (Figure 4). The results show that the deletion of gene *VM1G_06867* can compensate for the deficiency of gene *VmSom1* in the medium containing Congo red, NaCl, and Sorbitol. But it fails to completely remedy this defect.

### 3.5. Impact of VM1G_06867 on Utilization of Carbon and Nitrogen Sources

We incubate all strains on Czapek’s medium containing different carbon sources to assay the effect of gene *VM1G_06867* on the utilization of various carbon sources. Lactose, trehalose, fructose, maltose, and glucose of equal quality, respectively, are used in place of sucrose. The results indicate that the mycelial growth rate of the *VM1G_06867* single deletion mutant is significantly higher than that of the wild-type strain in sucrose and fructose and significantly less than that of the wild-type strain in lactose and glucose. There is no difference in the growth rate of mycelium in other carbon sources. The analysis of carbon source utilization in the gene double deletion mutants reveals that the hyphal growth rate of *ΔVmSom1/06867* is significantly lower than that of the *VmSom1* single deletion mutant in the medium containing lactose and fructose. The growth rate of hyphae in a sucrose medium is significantly higher than that of the *VmSom1* single deletion mutant (Figure 5).

Moreover, to assay the effect of the *VM1G_06867* gene on the utilization of different nitrogen sources, we incubate all strains on Czapek’s medium containing different nitrogen sources. NaNO_3_ is replaced with peptone, urea, KNO_3_, and (NH_4_)_2_SO_4_ of equal quality, respectively. The results of nitrogen source utilization analysis show that the mycelial growth rate of the *VM1G_06867* single deletion mutant is significantly higher than that of the wild-type strain in peptone, KNO_3_, and (NH_4_)_2_SO_4_. When using multicomponent complex nitrogen sources such as organic nitrogen peptone, the *VM1G_06867* single deletion mutant has the fastest mycelium growth rate. When using single-component nitrogen sources such as urea, the growth rate of mycelium in the *VM1G_06867* single deletion mutant is slower compared to sdau11-175. This indicates that the gene *VM1G_06867* has an effect on the utilization of nitrogen sources. Analysis of nitrogen source utilization in the gene double deletion mutants reveals that the growth rate of *ΔVmSom1/06867* in the medium containing peptone is significantly higher than that of wild-type strain 11-175 and the *VmSom1* single deletion mutant (Figure 5).

### 3.6. Absence of VM1G_06867 in Pathogenicity

We perform pathogenicity tests to determine whether *VM1G_06867* is involved in disease transmission. We inoculate mycelial plugs of various strains onto apple twigs and fruit. After inoculating for 9 days, it was observed that the lengths of the lesions on the apple fruit with *VM1G_06867* deletion mutant inoculations had decreased by approximately 10.8% compared to those with the WT (Figure 6B). It was also found that the lengths of the lesions on the apple twigs were reduced by approximately 12.6% (Figure 6C). These results indicate that gene *VM1G_06867* is involved in the pathogenic process of *V. mali*. Compared with 11-175, the double deletion mutant *ΔVmSom1/06867* reduces the diameter of apple fruit spots by 79.9%, whereas the *VmSom1* single deletion mutant increases the diameter of apple fruit spots by 75.9% (Figure 6). The above results show that the deletion of gene *VM1G_06867* can compensate for the deficiency of reduced pathogenicity caused by the deletion of *VmSom1* but cannot completely compensate.

## 4. Discussion

*VM1G_06867* is a Zn2Cys6-type zinc finger transcription factor. The zinc finger transcription factor family consists of transcription factors widely found in eukaryotes [22,23]. Zn2Cys6-type transcription factors are not highly conserved, and their function varies greatly, being involved in the regulation of different functions in different fungi [24]. These transcription factors have important regulatory effects on the mycelial growth rate, type, and morphology, spore morphology and quantity, metabolite formation, stress response, and pathogenicity. Currently, more in-depth studies have been conducted on the homologous gene *flo8* (*S. cerevisiae*) of the *Som1* gene, including studies in filamentous fungi such as *A. nidulans*, *M. oryzae*, *V. dahliae*, *A. fumigatus*, and *M. acridum* [13,14,15,16].

In this study, we identify *VM1G_06867* in *V. mali*. The results show that *VM1G_06867* performs crucial roles in growth, pathogenicity, asexual development, and the maintenance of cell wall integrity. *VM1G_06867* restores osmotic stress and cell wall integrity defects caused by the deletion of *VmSom1*, and it also restores the loss of pathogenicity caused by the deletion of the *VmSom1* gene but not completely. *VM1G_06867* is a C6-type zinc finger protein transcription factor of *V. mali* that affects the growth of *V. mali* hyphae. The results are consistent with those of *Magnaporthe grisea* [9,25].

It is well established that conidial reproduction is important for fungi to survive in nature, especially as *V. mali* mainly infects the host bark via conidia. In this study, conidiation capacity increases in the *VM1G_06867* deletion mutant, which shows that *VM1G_06867* negatively regulates conidiation, similar to the findings in *P. marneffei* [26]. In addition, in *A. parasiticus* and *A. flavus*, the msnA proteins act as negative regulators of conidiation [27]. Although the deletion of *VM1G_06867* slows conidial germination, aberrant conidia are not observed. In *A. brassicicola*, *Abvf19* is independent in conidial germination [28].

Furthermore, *VM1G_06867* is involved in the pathogenic process of apple putrefaction in this study. However, in *Fusarium graminearum*, *EBR1*(enhanced branching 1) affects the polar growth and virulence of the pathogen. Detailed analysis shows that the reduced radial growth might be due to reduced apical dominance of the hyphal tip, leading to increased hyphal branching, and preventing the hyphal tip from puncturing the host cell [8]. In addition, Chambers et al. found that the Zn (II) _2_Cys_6_ -type transcription factor *ptf1* of *Leptosphaeria maculan* weakened the pathogenicity via base pair substitution during in vitro passage [29]. In *Valsa mali*, the deletion of *VM1G_06867* slows conidial germination, which may be one of the reasons for the reduced pathogenicity.

*VM1G_06867* affects the cell wall integrity of the apple rot fungus, resulting in a significant decrease in the tolerance of mutants to cell wall growth inhibitors such as SDS, CFW, and Congo red. Studies have shown that Congo Red and CFW exert significant inhibitory effects on deletion mutants *ΔmidA* and *Δrom2* in *A. fumigatus* [30]. Similarly, the tolerance of mutants to cell wall synthetic inhibitors decreases significantly after *MAPK slt2* gene knockout in Candida glabrata [31]. These changes are related to the cell wall integrity pathway (CWI). However, the cell wall growth inhibitor also has a significant inhibitory effect on the knockout mutant of the *VM1G_06867* gene. Further investigation is needed to understand the role of the *VM1G_06867* gene in the CWI pathway in the future.

In general, the regulation of growth and development by transcription factors is multilevel and multifaceted, and changes in a single phenotype may be controlled by multiple transcription factors, while a single transcription factor may also affect multifaceted phenotypic changes. Each TF gene regulates the expression of many downstream genes, and each gene expression is regulated by many TF genes so that each mutant phenotype is controlled via the expression changes of several genes [9].

## 5. Conclusions

In summary, we constructed *VM1G_06867* single deletion mutants and double deletion mutants. *VM1G_06867* plays important roles in growth, asexual development, maintenance of cell wall integrity, and carbon source utilization. *VM1G_06867* can recover osmotic stress and cell wall integrity defects caused by the deletion of *VmSom1*, as well as restore the loss of pathogenicity caused by the deletion of the *VmSom1* gene, but not completely. Our research provides clear evidence that testifies to the molecular pathogenic mechanism of *VM1G_06867* in *V. mali*.

## Figures and Tables

**Figure 1 jof-09-00692-f001:**
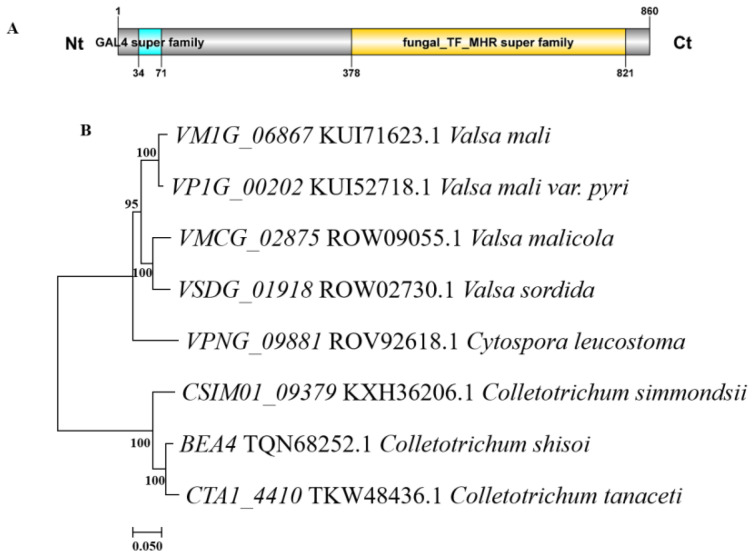
Domain structures and phylogenetic analysis of *VM1G_06867*. (**A**) The gene sequence is 3228-bp and encodes a protein of 860 amino acids (AA). The VM1G_06867 protein contains a GAL4 super family domain at 34–71AA. Additionally, a fungal_TF_MHR super family domain is predicted at amino acids 373–821 of VM1G_06867 protein. (**B**) Phylogenetic analysis of VM1G_06867 of *Valsa mali* and its homologs in other fungi. The VM1G_06867 proteins are analyzed using MEGA 7 and neighbor-joining analysis with 1000 bootstrap replicates. The numbers on the branches represent the percentage of replicates supporting each branch. The bar represents a divergence of 20% in the sequence.

**Figure 2 jof-09-00692-f002:**
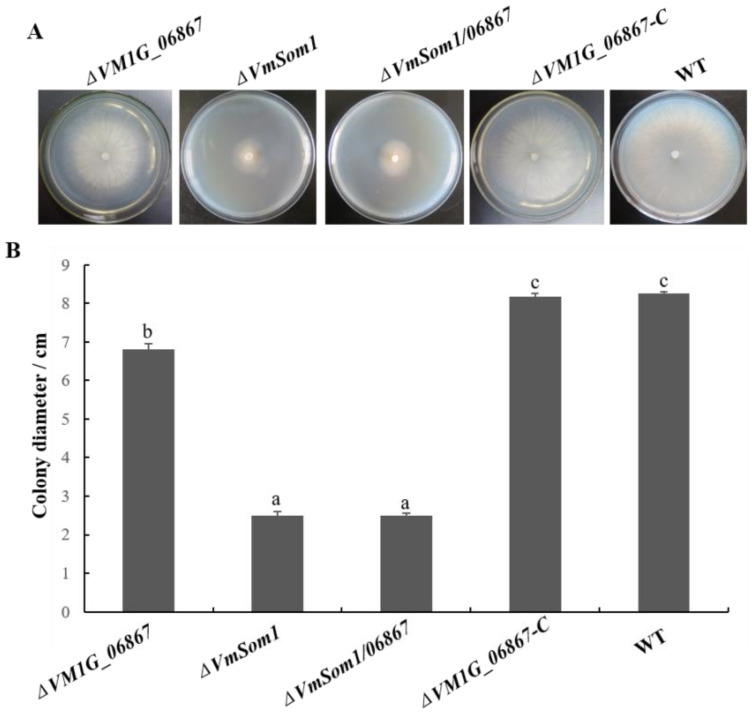
Effects of deletion of *VM1G_06867* on mycelial growth in *Valsa mali*. (**A**) Colony morphologies on PDA after growth for 3 days. (**B**) Measurement of colony diameter after 3 days. The mean and standard deviation, which are calculated using data from three biological replicates. The statistical analysis is performed using Fisher’s least significant difference (LSD) in the SPSS software package. Different letters indicate statistically significant differences (Duncan’s new multiple range test, *p* < 0.05).

**Figure 3 jof-09-00692-f003:**
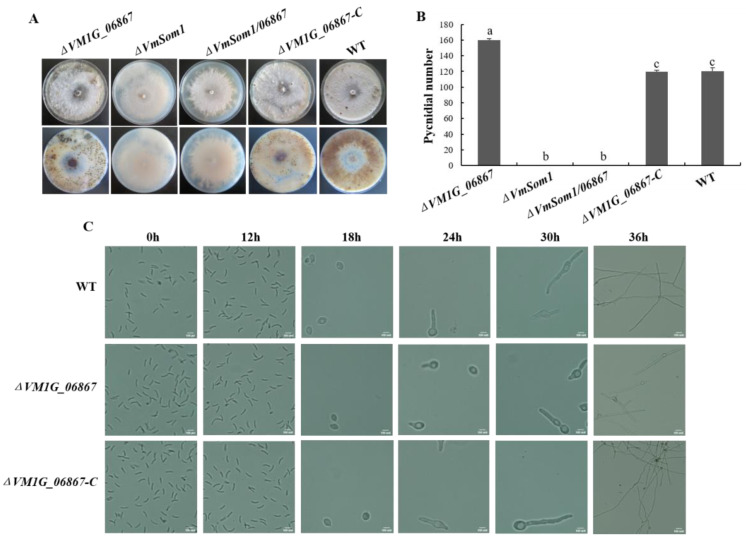
Effects of deletion of *VM1G_06867* on conidiation and conidia germination in *Valsa mali*. (**A**) Growth of pycnidia in wild type, *VM1G_06867* single deletion mutant, complemented mutant, and double deletion mutant on PDA for 20 days PDA. (**B**) Quantification of produced pycnidia produced as described in Section 2. Data from three replicates are analyzed using the protected Fisher’s least significant difference (LSD) test. Different letters indicate statistically significant differences (Duncan’s new multiple range test, *p* < 0.05). (**C**) Coating of spore suspensions from different strains onto PDA media at 25 °C for 30 h. Bars = 10 μm.

**Figure 4 jof-09-00692-f004:**
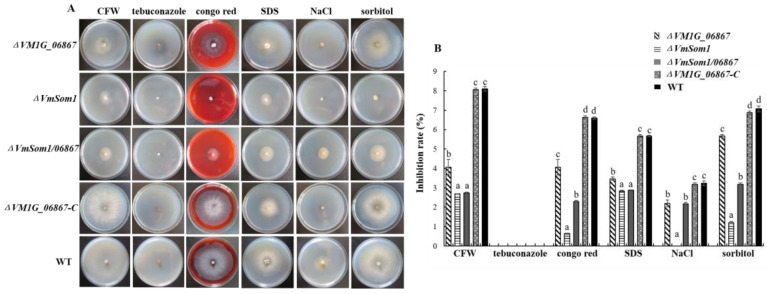
Effects of deletion of *VM1G_06867* on stress responses in *Valsa mali*. (**A**) The wild type, *VM1G_06867* single deletion mutant, complemented mutant, and double deletion mutant are inoculated on PDA supplemented with NaCl, sorbitol (osmotic stress), Congo red, fluorescent brightener (cell wall inhibitor), tebuconazole (germicide), and SDS. Images are taken after 3 days on different stress media. (**B**) The mean and standard deviation, which are calculated using data from three biological replicates. The Fisher’s least significant difference (LSD) in the SPSS software package is used for statistical analysis. Different letters indicate statistically significant differences (Duncan’s new multiple range test, *p* < 0.05). Bars indicate standard deviations of the mean of the three replicates.

**Figure 5 jof-09-00692-f005:**
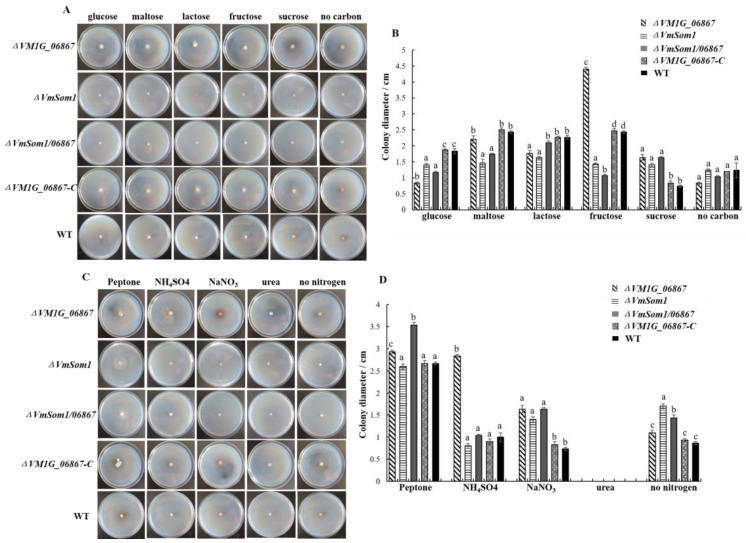
Effect of carbon and nitrogen sources on hyphal growth of gene *VM1G_06867*. (**A**) All strains are incubated on Czapek’s medium containing different carbon sources, respectively. Sucrose is replaced with lactose, fructose, maltose, and glucose of equal quality, respectively. (**B**) The mean and standard deviation, which are calculated using data from three biological replicates. The Fisher’s least significant difference (LSD) in the SPSS software package is used for statistical analysis. Different letters indicate statistically significant differences (Duncan’s new multiple range test, *p* < 0.05). Bars indicate standard deviations of the mean of the three replicates. (**C**) All strains are incubated on Czapek’s medium containing different nitrogen sources, respectively. NaNO_3_ is replaced with peptone, urea, KNO_3,_ and (NH4)_2_SO4 of equal quality, respectively. (**D**) The mean and standard deviation, which are calculated using data from three biological replicates. The Fisher’s least significant difference (LSD) in the SPSS software package is used for statistical analysis. Different letters indicate statistically significant differences (Duncan’s new multiple range test, *p* < 0.05). Bars denote standard deviations of the mean of the three replicates.

**Figure 6 jof-09-00692-f006:**
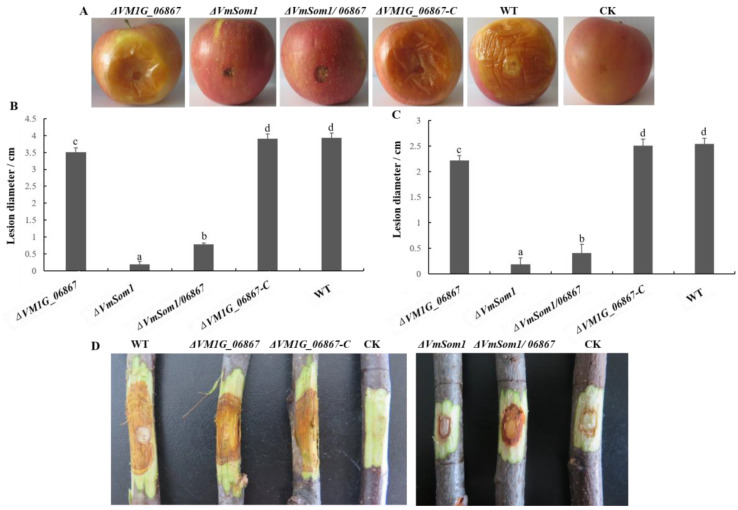
Effects of *VM1G_06867* on pathogenicity in *Valsa mali*. (**A**) Apple fruit is scalded and inoculated with mycelium agar plugs from the wild type, *VM1G_06867* single deletion mutant, complemented mutant, and double deletion mutant. (**B**) The lesion length that is measured at 7 days post inoculation (dpi). The experiments are repeated three times. Different letters indicate statistically significant differences (Duncan’s new multiple range test, *p* < 0.05). (**C**) Apple twigs are scalded and inoculated with mycelium agar plugs from the wild type, *VM1G_06867* single deletion mutant, complemented mutant, and double deletion mutant. Images are taken at 7 dpi. (**D**) The lesion length that is measured at 7 dpi. The experiments are repeated three times. Different letters indicate statistically significant differences (Duncan’s new multiple range test, *p* < 0.05).

## Data Availability

All the data that support the findings of this study are available in the paper and its Appendix A published online.

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
