# Peer review of "Transcription Factor VM1G_06867: A Requirement for Growth, Pathogenicity, Development, and Maintenance of Cell Wall Integrity in Valsa mali"

_jof, 2023, doi:10.3390/jof9060692_

Round 1

Reviewer 1 Report

In the present study by Diao et al, the authors investigated the role of the transcription factor Vm1G-06867 in regulating the growth and virulence of the fungal pathogen Valsa mali. The authors use Vsom1 deletion mutant strain, an important transcription factor that has been previously implicated in regulating growth and virulence of several fungi. The authors used the Vsom1 deletion strain to identify potential genes that may be implicated in regulating fungal virulence in the absence of Vsom1. They identified Vm1G-06867 as a differentially expressed gene in a transcriptomic analysis between of Wt and Vsom1 deletion mutant, however this analysis is not presented in the study. The authors perform deletion and complementation studies of Vm1G and identify that this gene plays a role in growth and conidiation of V. mali and to be implicated in stress responses. Finally, the authors perform virulence studies of Vm1G in apple fruit and twigs and demonstrate that Vm1G partially restores the pathogenicity in the Vsom1 deletion strain. Overall, the studies are logical with deletion and complementation strategies used to describe the role of Vm1G-06867 in V. mali. However, the studies are largely descriptive and unclear as to how Vsom1 regulates Vm1G-06867 and how Vm1G-06867 promotes virulence in the absence of Vsom1.

Comments:

1.    The authors identified Vm1G_06867 in a transcriptomic analysis, this is an important piece of data that needs to be presented in the manuscript. It is not clear why the authors do not present this data and rather they mention it as unpublished. What kind of transcriptomic data did the authors perform, was it RNASeq or other transcriptomic data?

2.     What were the other genes/gene network that were differentially expressed in the Vsom1 deletion? Were there other Vm1G proteins also effected in the Vsom1 deletion such as Vm1G-1794?

3.    The Virulence studies done in apple fruit and twigs are confusing and not clear. Firstly, the single deletion Vm1G strain was associated with a marginal loss in its pathogenicity in agreement with their in vitro growth defect observed in Fig3. Paradoxically, in the double mutant (Vm1G/Vsom1), Vm1G deletion partially enhanced virulence. The authors do not provide any mechanistic insights as to how Vm1G is regulating virulence in the absence of Vsom1.

4.    A recent study (PMID: 36449354) knocking out Vm1G-1794 in V. mali found that Vm1G-1794 regulated autophagic degradation of virulent factors to promote apple canker, this is an important study in the apple canker field and needs to be cited. It may be interesting if a similar mechanism is operational in the Vsom1 deletion strain, did the authors evaluate the autophagic responses?

Author Response

请参阅附件

Reviewer 2 Report

The authors identify a gene in the fungus Valsa mali that possibly codes for a binuclear zinc cluster transcriptional factor and investigate its role in the virulence of this fungus in apple and the effect of its deletion on tolerance or resistance to changes in the carbon and nitrogen source and other compounds and cations added to the culture medium.

The objective of the work is correctly developed experimentally. The conclusions obtained are based on the growth tests of the mutant strains generated in culture media in which a series of compounds have been modified or added to allow the basic evaluation of the metabolism and homeostasis/tolerance of this fungus to these compounds and nutritional sources.

The main criticism is the poor description of the transcriptional factor both at the level of structure and possible interpretations of the evolutionary conservation of its function, as a very basic comparative/phylogenetic study is developed. In fact, the phrase .We predict two conserved zinc finger DNA-binding domains at 34–71 and 373–821 AA. (Fig 1A) ”  is erroneous. This transcriptional factor has a DNA binding domain that is perfectly characterised in ascomycete fungi and is a binuclear zinc cluster and the domain at the C-terminal location is characteristic of this type of factors but it is not a DNA binding domain.

The authors need to expand on the results section 3.1.

Instead, sections 3.2 and 3.3 need to be merged. A section 3.2 dedicated to the construction and verification of the construction of the mutant strains does not make sense. They can be described in the materials and methods, and figure 2 can be added to figure 3 or moved to the supplementary material.

It is also important to explain correctly the logic of using the som1 null allele. Figure 4 already shows images of this strain but all this material is not connected to the description of the mutant phenotypes. In fact, on line 235, without prior introduction, the phenotype of the double null mutant strain VmSom1/0686 starts to be described. This section needs reorganisation and improvement.

The discussion needs more elaboration. I suggest the authors improve on integrating the results into previous knowledge and elaborate hypotheses on the mode of action of this TF.

Please proofread the English and correct several typos.

Round 2

Reviewer 1 Report

I have gone through the revised manuscript and the response to my comments. The authors have clarified the comments and the manuscript reads better in its present form.